# Central Nervous System Disorders with Auto-Antibodies in People Living with HIV

**DOI:** 10.3390/microorganisms12091758

**Published:** 2024-08-24

**Authors:** Giacomo Stroffolini, Cristiana Atzori, Daniele Imperiale, Mattia Trunfio, Giovanni Di Perri, Andrea Calcagno

**Affiliations:** 1Department of Infectious-Tropical Diseases and Microbiology, IRCCS Sacro Cuore Don Calabria Hospital, Via Don Angelo Sempreboni 5, Negrar, 37024 Verona, Italy; 2Unit of Neurology, Ospedale Maria Vittoria, ASL Città di Torino, Via Luigi Cibrario, 72, 10144 Torino, Italy; domp@aslto2.piemonte.it (C.A.); daniele.imperiale@aslcittaditorino.it (D.I.); 3Unit of Infectious Diseases, Department of Medical Sciences, University of Torino, Ospedale Amedeo di Savoia, C.so Svizzera 164, 10124 Torino, Italy; mattia.trunfio@edu.unito.it (M.T.); giovanni.diperri@unito.it (G.D.P.); andrea.calcagno@unito.it (A.C.)

**Keywords:** HIV, neuroimmunology, neurovirology, neuroinflammatory diseases, neuroinfectious diseases, CNS-antibodies

## Abstract

**Abstract:** People living with HIV (PLWH) may present atypical neurological complications. Recently, autoimmune manifestations of the central nervous system (CNS) have been described. We retrospectively described the features of PLWH presenting with acute neurological symptoms with positive anti-CNS antibodies. We analyzed relevant CSF characteristics. Twelve patients were identified, with demyelinating, inflammatory, or no MRI lesions. We observed CSF inflammatory features. Aspecific CSF anti-CNS antibodies were found in all subjects and a specific antibody (second-level blotting panel) was found in one. The cases presented a slow resolution of symptoms with sequelae. More studies are needed to better describe the spectrum and prognosis of autoimmune CNS diseases in PLWH.

## 1. Introduction

People living with HIV (PLWH) may present complications after the introduction of combined antiretroviral therapy (cART), including cardiovascular, metabolic, and neurological issues. HIV encephalitis is currently rare but other neurological disorders are not uncommon, including HIV-associated neurocognitive disorder (HAND), which is now estimated to be as prevalent as 20–30% in PLWH with controlled HIV viremia. While these disorders have a complex pathogenesis, chronic inflammation may also play an important role; abnormal levels of cerebrospinal fluid (CSF) neopterin have been shown even after years of successful antiviral treatment [1,2]. Additionally, autoimmune disorders of the central nervous system (CNS) have been reported in PLWH, and the spectrum now includes anti-NMDAR encephalitis and anti-NSA encephalitis with psychotic onset [3,4,5,6]. Immune response inflammatory syndrome (IRIS) may occur after the start of antiretroviral treatment; recently, a new, uncommon acute CNS inflammatory condition has been described and reported to be related to the brain parenchyma invasion of CD8+ T lymphocytes [7,8]. On the contrary, PLWH with multiple sclerosis have a mild course and a less aggressive disease, suggesting that immune modulation may be beneficial in the context of immune-mediated disorders [9].

Due to loss of immune competence, inflammatory and autoimmune diseases can occur in PLWH, in whom diagnosis is more difficult due to differences in classical antibody titer and onset [10]. Immune system dysregulation is one of the key recognized features of the pathogenesis of non-AIDS-related comorbidities, and recently, autoimmune manifestations of the central nervous system (CNS) have been described. HIV patients have a higher incidence of systemic autoimmune diseases, even on highly active antiretroviral therapy (HAART) [11], commonly with broad spectrum production of non-organ specific antibodies and hypergammaglobulinemia [12]. T_reg_ cells and Th_17_ cells may also play a role in HIV infection and progression to AIDS and the development of autoimmune diseases. [13]. Additionally, other mechanisms have been proposed: HIV exerts a direct effect on B cells through its binding to B cell surface molecule CD40. Molecular mimicry between HIV proteins and self-antigens could cause antibody cross-reactions and therefore lead to broad autoimmune manifestations [14]. Given the limited number of reported cases and the potential underdiagnosis of these disturbances, we aimed to describe the clinical and diagnostic features of PLWH presenting autoimmune disorders of the CNS.

## 2. Materials and Methods

We retrospectively reviewed cases of PLWH with CNS disorders who had been admitted to the Unit of Infectious Diseases, Amedeo di Savoia Hospital, Torino (Italy), between 2001 and 2019 and presented anti-CNS auto-antibodies. The patients signed a written informed consent form for CSF withdrawal, storage, and analysis. The study was approved by the local Ethics Committee (Comitato Interaziendale di Orbassano, “PRODIN study”, protocol number 103). We included patients with confirmed positivity for serum/CSF anti-CNS antibodies identified using immunofluorescence and immunoblot methods (Euroimmun AG PerkinElmer company (Waltham, MA, USA)). Any patient with active central nervous system infections, such as acute confirmed infectious encephalitis of any etiology, mycobacteriosis, herpes viruses (beyond just isolated detection of EBV DNA), neurosyphilis, or cryptococcosis, was excluded from the study. Any patient with a known autoimmune disease was excluded from the study. EBV DNA was tested when possible for its role in neuro-inflammatory diseases. Beyond first-level CSF markers (cells, proteins, and glucose ratio), CSF was tested for blood–brain barrier (BBB) permeability (CSF to serum albumin ratio, “CSAR”), inflammation (CSF to serum IgG ratio, neopterin), amyloid deposition (1–42 β-amyloid), neuronal damage (Total tau (T-tau)), phosphorylated tau (P-tau), 14-3-3 protein), and astrocyte damage (S-100 β). Quantitative determination of albumin in serum and CSF was measured using immunoturbidimetric methods (AU 5800. Beckman Coulter, Brea, CA, USA); 14-3-3 protein was detected using the Western blot (WB) technique; and CSF tau, P-tau, and 1–42 β-amyloid were measured using immunoenzymatic methods (Fujirebio diagnostics, Malvern, PA, USA). Neopterin and S-100β were measured through validated ELISA methods [DRG Diagnostics (Marburg, Germany) and DIAMETRA S.r.l. (Spello, Italy), respectively]. Imaging (either MRI or CT) and electrophysiological studies (EEG) were performed. Data were analyzed using standard statistical methods: variables were described as numbers (percentages) with medians [interquartile ranges (IQR) or ranges (minimum–maximum)]. Due to limited power, further subgroup analysis was not feasible. Data analysis was performed using SPSS software for Mac (version 26.0. IBM Corp (Armonk, NY, USA)).

## 3. Results

We identified twelve patients: they were more frequently male (75%) with a median age of 51 years [range: 42–55], 75% had European ancestry, and 50% took HAART. The median and nadir CD4 cell count were 302/mm^3^ [range: 120–434] and 93/mm^3^ [range: 18–259]; median plasma and CSF HIV RNA were respectively 922 [range: 0–31,556] and 2765 copies/mL [range: 32–21,829]. A median of 4 years [range: 1–7] since the diagnosis of HIV infection was recorded. When considering the MRI results, the participants presented with demyelinating (n = 3, 25%) or inflammatory lesions (2, 17%) while one (8%) presented with brain atrophy; a mixed pattern was observed in three subjects (25%) while the remaining three individuals showed a normal MRI (25%) at the onset of symptoms. Additional clinical and radiological features of the participants are presented in Table 1 and the Appendix A.

Symptoms included headache, ataxia, dizziness, shaking, and agitation as well as walking, fluency, and memory disorders. CSF was clear in all patients with a median of 10 cells/mm^3^ [0–40], 65 mg/dL of proteins [41–126], and 52 mg/dL of glucose [42–65]. Median CSF neuro-markers were as follows: T-tau 160 pg/mL [87.5–362.2], P-Tau 29.5 pg/mL [19.2–37.2], neopterin 2 ng/mL [1–12.2], S100 Beta 120.5 pg/mL [52.2–179.2], and Beta Amyloid_1–42_ 889 pg/mL [401.5–1189.2]. CSF 14-3-3 protein was positive in three patients (25%). Increased BBB permeability was observed in six participants (50%). No patient tested positive for any common pathogen responsible for meningoencephalitis, as per the inclusion criteria. In two patients in which EEG was studied, a diffuse cortical wave reduction was observed. Broad CSF anti-CNS antibodies (immunofluorescence technique) were found in all subjects and only one patient tested positive for specific antibodies (Recoverin, Zic4) when using the second-level blotting panel. The pattern of immunofluorescence was nuclear in five patients, synaptic in three, synaptic and nuclear in one, and non-specific in three (Table 2, Figure 1, Figure 2 and Figure 3 for negative and positive controls).

All but one patient had both serum and CSF antibody positivity. In the majority of cases, we observed a slow resolution of symptoms with sequelae (67%); specifically, five patients experienced residual sensory disorders, and three patients experienced neuropsychological (mainly memory) disorders. Two cases of concomitant symptomatic CSF escape were registered with complete resolution of symptoms after HAART optimization (Table 1). In patients 4, 5, and 8, CSF, EBV, and DNA were 54, 82, and 315 cp/mL, respectively (Table 1), and negative in the other four patients in which they were assessed (patients 6, 7, 11, and 12).

## 4. Discussion

In this case series, we measured several cerebrospinal fluid biomarkers of neuronal damage and inflammation of neurologically symptomatic PLWH. Our findings are consistent with classical CSF inflammatory features suggestive of encephalitis. More widely, our series may represent a good example of the variety of clinical manifestations in HIV-associated neurological disorders, moving through different pathological biotypes, as proposed in a recent study [15]. CSF biomarkers were altered in several patients with non-homogenously distributed patterns, yet they were consistent with disruption of the BBB, neuronal damage, and inflammation. There were no other factors favoring autoimmune disorders that may explain these features and no cases of para-neoplastic diseases were recorded; the role of EBV in such alterations remains uncertain. All patients had CNS-antibody positivity confirmed in CSF and all but one in plasma. We were not able to determine a specific antibody pattern to any of the standard antigens except for one case (patient n. 12 in Table 1) in which Recovering and Zic4 antibodies were found; given the association with neoplasias and the higher risk observed in PLWH, this subject is on active follow-up [14,16]. The majority of our patients had altered imaging with a high proportion of demyelinating and mixed lesions upon brain MRI; while HIV infection per se has been associated with such patterns, it would be interesting to understand whether auto-antibodies are the cause or consequence of neuronal damage. Viral infections have been described as potential triggers for several autoimmune disorders including encephalitis (the commonest being anti-nMDAR encephalitis after HSV CNS infections) [17]. It is also worrisome that several patients had neurological sequelae and this should prompt the early diagnosis and treatment of patients showing unexpected course or poor response to antiretroviral therapy. To our knowledge, few examples of such a correlation have previously been described, making this an important field for future research, especially considering the high prevalence of symptoms of such presentations and the current difficulty in reaching a reliable diagnosis. It is noteworthy that all of these patients presented with acute onset and neurological symptoms that could not be recognized using prompt classical classification and diagnostic tools (first-line assessment). A possible role of human herpesviridae is under investigation as a factor that leads to the progression of autoimmune encephalitis; HIV could actively have a similar role due to unique characteristics such as latency and its reversion, DNA total viral load, chronic microglial activation, and CNS inflammation in which other non-AIDS-defining conditions add load to the total pathological burden and promote the auto-inductive mechanism of progression to neurological issues that clinicians encounter daily [18,19]. This has been largely proven for HAND and aspecific conditions such as white matter hyperintensities, but the field of autoimmunity has not been explored thoroughly [20,21]. The occurrence of aspecific auto-antibodies in patients with symptomatic CSF escape may support the role of HIV replication as a trigger for immune-mediated self-damage [5,20].

A possible role for endogenous human viruses and for persistent herpes viruses has been highlighted repetitively for many inflammatory conditions and Multiple Sclerosis (MS) is just one of them [22,23]. EBV, which was positive in three patients from our series, may manifest complex immune patterns that are not unique to this setting. Lentiviruses can induce a wide range of immune responses and lead to the development of antibodies, which have been tested in animal models. Autoimmune CNS disorders are largely underdiagnosed, and despite many reports describing autoimmunity spectra in PLWH, none of them include CNS complications. To our knowledge, few cases of autoimmune anti-CNS anti-NMDAr acute encephalitis have been described in PLWH, while only one case of anti-NSA+ encephalitis with psychotic onset has been reported [3,4,5,6]. NMDAR hyperactivation has been described in HIV infection, and gp120, a major component of HIV virus, may trigger NMDAr’s dephosphorylation [24]. Reports suggest that AE may be triggered by (or superposed to) herpes simplex infection which is common in such conditions, and the role of EBV is still under debate with some evidence of raised inflammation in its presence [25]. Other possible pathogenetic mechanisms are chronic immune CNS activation, the role of other pathogens in enhancing auto-antibody production, and the presence of a leaking BBB, as already discussed.

Several implications may deserve further discussion given the general high proportion of undiagnosed encephalitis. Classical findings help in the diagnostic process, but up to 30–40% of encephalitis cases remain of unknown origin despite considering autoimmune encephalitis (AE) as a major differential diagnosis, and AE may be more common than viral encephalitis when considering specifying settings [26,27]. Additional explanations could relate immune activation to viral products not normally found in the CSF of patients with acute disease or, looking further, potentially a large number of microorganisms could be reclassified as pathogens if a clear relation between their presence and development of acute symptoms were ever proven. New frontiers such as metagenomics may be helpful in combination with a standard strategy to improve highlighting credible etiological pathogens, but experience is still poor, and frequently, results of uncertain significance endure [28]. Nevertheless, it remains an attractive tool whose use is nowadays difficult to access due to cost and availability. To date, we lack working and reliable markers to test for additional diagnostics.

Moreover, it is important to stress how crucial this differential diagnosis could be in the absence of the recognition of classical pathogens and implications for early treatment at large, following the example of other inflammatory diseases and the potential impact on neurological sequelae when a more aggressive diagnostic path and management are delayed [29,30]. Immune dysregulation and chronic microglial activation, as already highlighted, can explain only part of these pathological patterns. The possibility of intervening promptly with immunomodulatory agents in order to ameliorate prognosis and sequelae may be attractive, specifically in cases where CD8+ or auto-antibody-mediated encephalitis is suspected [7,8,30].

This study has several limitations: its retrospective nature and the heterogeneity and diversity of symptom onset and patient characteristics, in particular with regard to treated and untreated patients. In some cases, uncertainty in the ultimate diagnosis remains. Moreover, we could not test myelin oligodendrocyte glycoprotein (MOG) antibodies. Finally, due to limited power, further subgroup analysis was not feasible. More studies are needed to better describe the spectrum and prognosis of autoimmune CNS diseases in PLWH

## 5. Conclusions

In conclusion, our series highlights the importance of considering autoimmune mechanisms in the management of PLWH presenting with acute and sub-acute CNS symptoms. Including unconventional autoimmune encephalitis among the causes of acute and subacute CNS manifestations should be considered in the diagnostic workup.

## Figures and Tables

**Figure 1 microorganisms-12-01758-f001:**
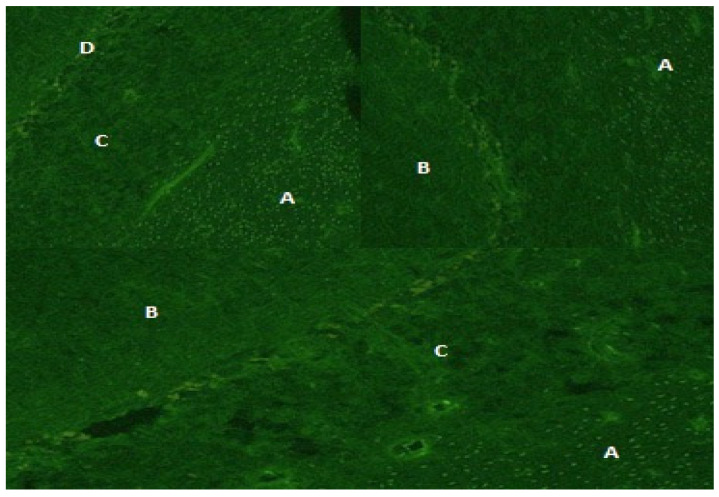
Anti-oligodendrocytes immunofluorescence on monkey cerebellar tissue samples exposed to the CSF of our patient (n. 5) collected at the onset of her symptoms. The diffuse positivity of the fluorescence among the white matter (A) and molecular (B) and granular layers (C) of the grey matter revealed the presence of anti-oligodendrocyte antibodies in the CSF of our patient. (Purkinje cells shown in D). This pattern is not specific.

**Figure 2 microorganisms-12-01758-f002:**
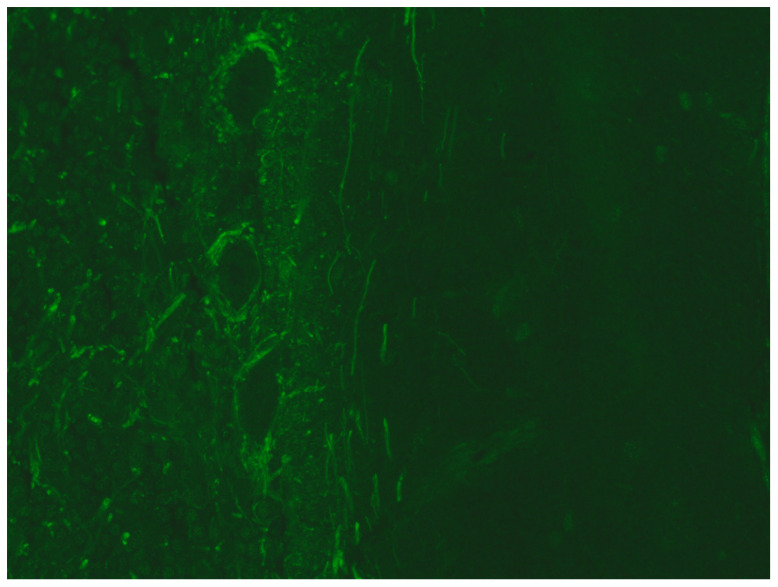
The figure demonstrates a perisomatic pattern, positive upon immunofluorescence for anti-CNS antibodies. Purkinje cells show fluorescence at the axonal level. Figure from patient case n. 8. This pattern is not specific.

**Figure 3 microorganisms-12-01758-f003:**
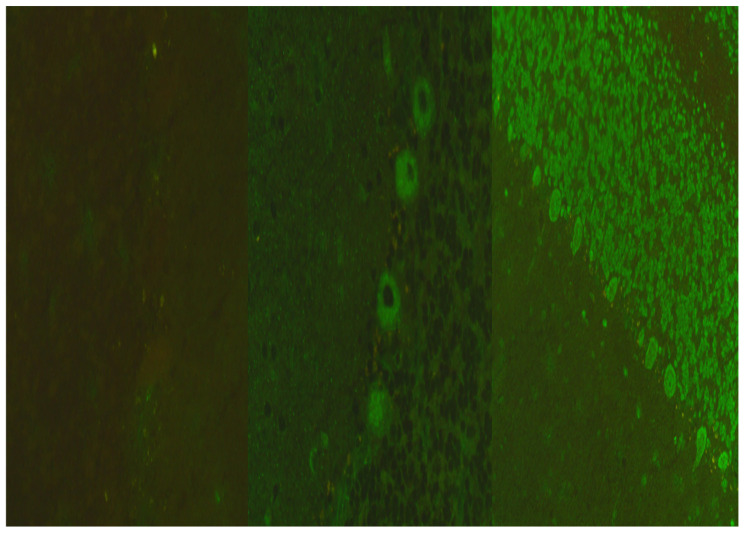
The figure on the left side demonstrates a negative control; the figure in the middle demonstrates a pattern for anti-Tr positivity: fluorescent in the cytoplasm of Purkinje cells. The figure on the right side demonstrates a pattern for anti-Hu positivity: fluorescent neuronal nuclei (both in the molecular layer, the granular layer, and Purkinje nuclei).

**Table 1 microorganisms-12-01758-t001:** Patients and main features. HIV: human immunodeficiency virus; CSF: cerebrospinal fluid; EEG: electroencephalography; HAART: highly active antiretroviral therapy; M: male; F: female; N/A: not assessed; BBBp: blood–brain barrier permeability; VDRL: venereal disease research laboratory test; CSAR: CSF to serum albumin ratio; PCNSL: primary central nervous system lymphoma; PML: progressive multifocal leukoencephalopathy; IRIS: immune reconstitution inflammatory syndrome.

Patient n°	Age	Gender	Imaging Pattern	Onset Symptoms	CD4+	HIV RNA Plasma (cp/mL)	HIV RNA CSF (cp/mL)	EBV DNA CSF (cp/mL)	CSF Ab Pattern	EEG	Altered Neuromarkers	HAART	Final Diagnosis
**1**	52	M	Demyelinating	Sensory and dysautonomic disorders	285	6679	4466	N/A	Synaptic	Normal	14.3.3, Neopterin, BBBp, CSAR, and IgG index	Treated	Myelitis
**2**	55	M	Normal	Asymptomatic, control in previous neurolue, and VDRL serum positivity	436	Negative	Negative	N/A	Nuclear	Normal	None	Naïve	None
**3**	28	M	Inflammatory/demyelinating	Confusion	251	16,197	1064	N/A	Nuclear	Normal	14.3.3, Tau, and Beta Amyloid	Naïve	Balo concentric sclerosis
**4**	60	F	Demyelinating	Sensory and extrapyramidal disorders	111	Negative	160	54	Nuclear and Synpatic	Normal	Intrathecal synthesis	Treated	PML-IRIS
**5**	50	F	Inflammatory/demyelinating	Extrapiramidal disorders and vertigo	516	Negative	7566	82	Aspecific	Normal	Intrathecal synthesis, CSAR, BBBp, Neopterin, and Beta Amyloid	Treated	CNS viral escape
**6**	41	M	Normal	Cefalea and sensory disorders	147	395	71	Negative	Nuclear	Normal	None	Naïve	HAART neurotoxicity
**7**	41	M	Demyelinating	Vertigo, sensory and extrapyramidal disorders, and ataxia	19	36,676	16,030	Negative	Synaptic	Normal	Neopterin, BBBp, and CSAR	Naïve	PML
**8**	44	F	Inflammatory	Dysautonomic and extrapuramidal disorders, confusion, and agitation	320	Negative	Negative	315	Aspecific	Altered	BBBp and intrathecal synthesis	Treated	None
**9**	60	F	Inflammatory	Movement disorders and ataxia	429	88,982	330,273	N/A	Aspecific	Altered	Intrathecal synthesis, BBBp, Neopterin, and Beta Amyloid	Treated	CSF viral escape
**10**	44	M	Demyelinating, inflammatory, and mass effect	Dysautonomic, sensory, and motor disorders	37	758,988	23,762	N/A	Synaptic	Normal	Tau, Neopterin, 14.3.3, and BBBp	Naïve	PCNSL
**11**	50	M	Atrophic	Extrapyramidal and movement disorders, ataxia, and confusion	379	Negative	Negative	Negative	Nuclear	Normal	None	Treated	None
**12**	55	M	Normal	Sensory disorders	715	1450	330,000	Negative	Nuclear, Recoverin, and Zic 4 on Blot	Normal	BBBp and Intrathecal synthesis	Naïve	None

**Table 2 microorganisms-12-01758-t002:** **Antibody patterns and neuromarkers.** In black, the patterns with corresponding neuromarkers outside the normal range of values are highlighted.

Altered Neuromarkers and Correspondances with Antibody Patterns	Nuclear	Synaptic	Aspecific	Zic4	Recoverine
**Neopterin**					
**1–42 Beta Amyloid**					
**BBB damage**					
**Intrathecal synthesis**					
**CSAR**					
**Tau**					
**FTau**					
**14.3.3.**					

## Data Availability

The data that support the findings of this study are available from the local Ethics Committee (Comitato Interaziendale di Orbassano, “PRODIN study”, protocol number 103). The datasets used and/or analyzed during the current study are available from the corresponding author upon reasonable request.

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
