# Peer review of "Central Nervous System Disorders with Auto-Antibodies in People Living with HIV"

_microorganisms, 2024, doi:10.3390/microorganisms12091758_

Round 1

Reviewer 1 Report

Comments and Suggestions for Authors

I consider that the study presented in the manuscript its interesting, however, I see it necessary that, although it is retrospective, the following information of the patients included be presented:

I suppose that in these twelve patients not only the markers described were determined: serum/CSF anti- CNS antibody, blood-brain-barrier (BBB) permeability, for inflammation (CSF to serum IgG ratio, neopterin),  amyloid deposition (1-42 β-amyloid), for neuronal damage, Total tau, Phosphorylated  tau, 14-3-3 protein and for astrocyte damage (S-100 β). I suppose that in the presence of the neurological manifestations indicated, the clinical and laboratory approach included the search for acute or chronic infectious pathologies that could occur in patients living with HIV, such as mycobacteriosis, herpes virus (I only identify the description of EBV DNA), neurosyphilis, cryptococcosis, etc.

It should have been pointed out that the presence of any active infectious pathology in the central nervous system such as those described above would be an exclusion criterion for this study?

It would even be desirable to explicitly indicate the history of opportunistic infections (CMV, mycobacteriosis, fungal infections, etc.) or co-infections (syphilis, HCV, HBV, HSV) of these twelve patients from the diagnosis of HIV infection. For now, it is unknown whether some of these infections may be involved in autoimmunity in patients living with HIV and this information would be useful, although the casuistry of the study is small.

It is also desirable to explicitly indicate whether any of the twelve patients had any diagnosis of systemic autoimmune disease prior to or concomitantly with HIV infection, which could be related to the presence of autoimmune brain damage.

Is this correction convenient on lines 83 and 84?: “mean plasma and CSF HIV RNA were respectively 922 [range 0-31556] and 2765 copies/mL [range 32-21829]”

On lines 84 and 85 is this correction appropriate?: “A median of 4 years [1-7] since the diagnosis of HIV infection was recorded”.

Comments on the Quality of English Language

I consider that the manuscript its written in an adequate english.

Author Response

Reviewer 1

I consider that the study presented in the manuscript its interesting, however, I see it necessary that, although it is retrospective, the following information of the patients included be presented:

I suppose that in these twelve patients not only the markers described were determined: serum/CSF anti- CNS antibody, blood-brain-barrier (BBB) permeability, for inflammation (CSF to serum IgG ratio, neopterin),  amyloid deposition (1-42 β-amyloid), for neuronal damage, Total tau, Phosphorylated  tau, 14-3-3 protein and for astrocyte damage (S-100 β). I suppose that in the presence of the neurological manifestations indicated, the clinical and laboratory approach included the search for acute or chronic infectious pathologies that could occur in patients living with HIV, such as mycobacteriosis, herpes virus (I only identify the description of EBV DNA), neurosyphilis, cryptococcosis, etc.

It should have been pointed out that the presence of any active infectious pathology in the central nervous system such as those described above would be an exclusion criterion for this study?

Response:

Dear Reviewer,

Thank you for your detailed and thoughtful feedback. In the presence of acute neurological manifestations, a comprehensive diagnostic workup was indeed conducted. This included screening for both first and second-level infectious pathogens. Any patient with active central nervous system infections, such as mycobacteriosis, herpes viruses (beyond just EBV DNA), neurosyphilis, or cryptococcosis, was excluded from the study. We have now clarified this more explicitly in the Methods section to ensure that the exclusion criteria are well understood.

Reviewer 1

It would even be desirable to explicitly indicate the history of opportunistic infections (CMV, mycobacteriosis, fungal infections, etc.) or co-infections (syphilis, HCV, HBV, HSV) of these twelve patients from the diagnosis of HIV infection. For now, it is unknown whether some of these infections may be involved in autoimmunity in patients living with HIV and this information would be useful, although the casuistry of the study is small.

It is also desirable to explicitly indicate whether any of the twelve patients had any diagnosis of systemic autoimmune disease prior to or concomitantly with HIV infection, which could be related to the presence of autoimmune brain damage.

Response:

Dear Reviewer,

Thank you for your valuable suggestions. We confirm that no patients (except two later diagnosed with PML, whose pathology is very peculiar, see table 1) had a history of central nervous system opportunistic infections, nor were they diagnosed with any autoimmune disease either prior to or concurrently with their HIV infection. We have now clarified this more explicitly in the Methods section. Additionally, we have provided details regarding the history of non-CNS opportunistic infections when present, as the Reviewer suggested (see supplementary table, that has been created to this end). Moreover, as shown in Table 1 and lines 87-91, this is an HIV population that has not been treated for a long time and has a relatively high nadir CD4 cell count, making opportunistic infections uncommon. This strengthens our findings, as it reduces the likelihood that any influence from other pathogens played a role

Reviewer 1

Is this correction convenient on lines 83 and 84?: “mean plasma and CSF HIV RNA were respectively 922 [range 0-31556] and 2765 copies/mL [range 32-21829]”

On lines 84 and 85 is this correction appropriate?: “A median of 4 years [1-7] since the diagnosis of HIV infection was recorded”.

Response

We thank the Reviewer for its contribution to the manuscript. We now amended the sentences as suggested.

Reviewer 2 Report

Comments and Suggestions for Authors

CNS disorders in HIV infection have been and still remain a significant problem, therefore, the topic presented is important from both: scientific and clinical point of view.  However, the above manuscript does not provide any significant new information referring to the autoimmune encephalitis, and the results and conclusions presentation are unclear.

Autoimmune encephalitis is not uncommon in HIV-negative patients. This was not addressed in the manuscript.

Twelve patients is a relatively small group (from how many patients they were selected?), subjects are very poorly described in the manuscript and the group is diverse, e.g. it is not acceptable to describe the HAART-treated and untreated subjects together.

Results of each subject should be presented individually, along with CNS marker test results.  Why was the presence of EBV tested (no description in Methods) and not other pathogens responsible for encephalitis?

The figures are of low quality. It is necessary to show the image of the positive immunofluorescence control.

The conclusion does not provide any scientific message.

Avoid the phrases: mostly, vast majority, etc.

Comments on the Quality of English Language

Must be improved. 

Author Response

Reviewer 2

CNS disorders in HIV infection have been and still remain a significant problem, therefore, the topic presented is important from both: scientific and clinical point of view.  However, the above manuscript does not provide any significant new information referring to the autoimmune encephalitis, and the results and conclusions presentation are unclear.

Autoimmune encephalitis is not uncommon in HIV-negative patients. This was not addressed in the manuscript.

Response

Dear Reviewer,

Thank you for your thorough evaluation of our manuscript. We fully agree on the critical importance of CNS disorders in HIV infection from both scientific and clinical perspectives. Our objective in this article is to shed light on the underexplored topic of autoimmune encephalitis in HIV-positive patients who present with acute neurological manifestations. Our work emphasizes the relevance of this condition within the HIV-positive population, an area that has been largely overlooked, with only individual case reports available to date.

We believe that this contribution is vital to raise awareness and improve suspicion of autoimmune encephalitis in HIV-positive patients, especially when they present with acute neurological symptoms.

To clarify, encephalitis itself is an uncommon disease in the general population, with an estimated incidence ranging from 3.5 to 7.4 per 100,000 person-years (https://doi.org/10.1007/978-981-99-6445-1_1; https://doi.org/10.1093/cid/ciz635). Autoimmune encephalitis represents only a modest subset of these cases. Despite growing awareness and better diagnostic capabilities, data indicate that the incidence of autoimmune encephalitis varies between 0.25 and 1 per 100,000 person-years, making it even rarer [ https://doi.org/10.1093/jalm/jfab102; doi:10.1016/S1474-4422(18)30244-8; https://doi.org/10.1212/WNL.0000000000001190].

Reviewer 2

Twelve patients is a relatively small group (from how many patients they were selected?), subjects are very poorly described in the manuscript and the group is diverse, e.g. it is not acceptable to describe the HAART-treated and untreated subjects together.

Response

Thank you for your thorough review of our manuscript. We acknowledge that the sample size of twelve patients is relatively small. In fact, the Editor made the proposition of changing our publication type to “Communication” instead of “Research Article”, in order to better reflect the nature of our study. These patients were selected from a larger cohort of approximately 5,000 individuals. We also appreciate your suggestion regarding the description of treatment status. In response, we have now clarified the data in the table 1 to distinguish between HAART-treated and treatment-naïve patients. Unfortunately, due to the limited number of patients, further subgroup analysis is not feasible. We have clarified all these limitations more explicitly in the revised manuscript in the dedicated section.

Reviewer 2

Results of each subject should be presented individually, along with CNS marker test results.  Why was the presence of EBV tested (no description in Methods) and not other pathogens responsible for encephalitis?

Response

EBV DNA was tested due to its known association with neuroinflammatory diseases. Additionally, all patients were tested for common encephalitis-causing pathogens. In particular, in the presence of acute neurological manifestations, a comprehensive diagnostic workup was indeed conducted. This included screening for both first and second-level infectious pathogens. Any patient with active central nervous system infections, such as mycobacteriosis, herpes viruses (beyond just EBV DNA), neurosyphilis, or cryptococcosis, was excluded from the study. We have now clarified this more explicitly in the Methods section to ensure that the exclusion criteria are well understood. Moreover, none of the patients had any opportunistic central nervous system infections or pre-existing/concomitant autoimmune diseases with HIV infection. These details, along with the patients' histories of non-CNS opportunistic infections when present, are now more clearly described in the Methods and Results section, and supplementary data (we have created an additional table to address this concern).

Reviewer 2

The figures are of low quality. It is necessary to show the image of the positive immunofluorescence control.

Response

We thank the reviewer for his suggestion. We now include a more high-quality iconography for both negative and positive controls (see supplementary data).  

Reviewer 2

The conclusion does not provide any scientific message.

Response

We respectfully disagree with the reviewer's assessment. However, we have revised the conclusion to improve clarity and to include a more informative scientific message.

Reviewer 2

Avoid the phrases: mostly, vast majority, etc.

Response

We thank the reviewer for his suggestions. We now amended the text avoiding such phrasing.

Reviewer 3 Report

Comments and Suggestions for Authors

Minor revision:

1. In fig.1 and fig.2 must include positive and negative control for comparison with your test results.

2. How about CSF infection ? for example; seropositive or microscopy of Cryptococcus which is most common in PLWH

3. Any data that support demyelination or inflammation?

4. How abouts auto-antibodies against multiple sclerosis?

Author Response

Reviewer 3

  1. In fig.1 and fig.2 must include positive and negative control for comparison with your test results.

Response

We thank the reviewer for his suggestion. We now included both positive and negative controls for comparison (see text and supplementary data).

Reviewer 3

  1. How about CSF infection ? for example; seropositive or microscopy of Cryptococcus which is most common in PLWH

Response

Thank you for your detailed and thoughtful feedback. In the presence of acute neurological manifestations, a comprehensive diagnostic workup was indeed conducted. This included screening for both first and second-level infectious pathogens. Any patient with active central nervous system infections, such as mycobacteriosis, herpes viruses (beyond just EBV DNA), neurosyphilis, or cryptococcosis, was excluded from the study. We have now clarified this more explicitly in the Methods section to ensure that the exclusion criteria are well understood.

Reviewer 3

  1. Any data that support demyelination or inflammation?

Response

Thank you for your question. The distinction between demyelination and inflammation in our study is primarily based on radiological findings and specific neuromarkers. We have now included a more detailed description in the text (and supplementary table that was created) to clarify this aspect. Additionally, the different patterns observed are summarized in Table 1 for easier reference.

Reviewer 3

  1. How abouts auto-antibodies against multiple sclerosis?

Response

Thank you for your insightful question. MOG antibodies were not tested in this study, and we have acknowledged this as a limitation in the manuscript. However, as the reviewer likely knows, given his/her expertise in this field, MOG antibody-positive patients represent only a subset of all multiple sclerosis cases. In our cohort, the clinical and marker patterns were not suggestive of MOG-related pathology, which is why testing for these antibodies was not pursued. This is now clarified in the limitations section.

Reviewer 4 Report

Comments and Suggestions for Authors

The manuscript: "Central Nervous System Disorders with Auto-antibodies in People Living with HIV" has thoroughly written introduction and adequately described methods. However, study group is quite miscellaneous and the number of participants involved is limited. The first concern is that a one half of the participants is on HAART (and usually with undetectable viral load as a result) and other half is not. The antibodies were detected in all patients but one, different neuromarkers are detected in patients with different diagnosis, and even though I understand that collecting data from participants that form a more similar group is not an easy task I feel this dataset would work better as a series of case reports. If the editor chooses to publish the manuscript I think the authors should at least address the possible differences of patients on therapy vs. untreated patients.

Comments on the Quality of English Language

The English language requires only minor interventions, resolvable during the editing process.

Author Response

Reviewer 4

The manuscript: "Central Nervous System Disorders with Auto-antibodies in People Living with HIV" has thoroughly written introduction and adequately described methods. However, study group is quite miscellaneous and the number of participants involved is limited. The first concern is that a one half of the participants is on HAART (and usually with undetectable viral load as a result) and other half is not. The antibodies were detected in all patients but one, different neuromarkers are detected in patients with different diagnosis, and even though I understand that collecting data from participants that form a more similar group is not an easy task I feel this dataset would work better as a series of case reports. If the editor chooses to publish the manuscript I think the authors should at least address the possible differences of patients on therapy vs. untreated patients

Response

Dear Reviewer,

We greatly appreciate your thoughtful insights and suggestions.

Following your recommendations, the editor has decided to change the publication format from a research article to a communication. This format better reflects the case series nature of our data and is more in line with your feedback.

We fully recognize the limitations of our study, particularly regarding the heterogeneous nature of our study group and the small sample size. While we understand the challenges this presents, we believe that our findings contribute important preliminary insights. We further disclose these limitations. In response to your concern about the differences between patients on HAART and those who are untreated, we have highlighted these distinctions in both the table and the text

Round 2

Reviewer 2 Report

Comments and Suggestions for Authors

The organization of Table 1 should be reversed, i.e. Pts No. in a vertical column.

Images of the positive and negative controls should be presented together (in one figure) with the tested clinical samples, not in the supplementary materials.

Comments on the Quality of English Language

No remarks

Author Response

Reviewer

The organization of Table 1 should be reversed, i.e. Pts No. in a vertical column.

Images of the positive and negative controls should be presented together (in one figure) with the tested clinical samples, not in the supplementary materials.

Response

We have accommodated the reviewer's requests. The organization of Table 1 has been reversed, with Pts No. now presented in a vertical column. Additionally, we have combined the images of the positive and negative controls with the tested clinical samples into a single figure, as requested, and removed them from the supplementary materials.